# Nanoscale covalent organic frameworks for enhanced photocatalytic hydrogen production

Wei Zhao [1], Liang Luo[1], Muyu Cong[2], Xueyan Liu[2], Zhiyun Zhang [2], Mounib Bahri [3], Boyu Li[1], Jing Yang[1], Miaojie Yu[1,2], Lunjie Liu [4], Yu Xia[4], Nigel D. Browning[3], Wei-Hong Zhu [2], Weiwei Zhang [2] ✉ & Andrew I. Cooper [1] ✉

Nanosizing confers unique functions in materials such as graphene and quantum dots. Here, we present two nanoscale-covalent organic frameworks (nano-COFs) that exhibit exceptionally high activity for photocatalytic hydrogen production that results from their size and morphology. Compared to bulk analogues, the downsizing of COFs crystals using surfactants provides greatly improved water dispersibility and light-harvesting properties. One of these nano-COFs shows a hydrogen evolution rate of 392.0 mmol $g^{-1}$ $h^{-1}$ (33.3 µmol $h^{-1}$), which is one of the highest mass-normalized rates reported for a COF or any other organic photocatalysts. A reverse concentration-dependent photocatalytic phenomenon is observed, whereby a higher photocatalytic activity is found at a lower catalyst concentration. These materials also show a molecule-like excitonic nature, as studied by photoluminescence and transient absorption spectroscopy, which is again a function of their nanoscale dimensions. This charts a new path to highly efficient organic photocatalysts for solar fuel production.

Reducing materials to the nanoscale is a powerful tool in materials science that can create properties that are not observed in bulk solids[1]. Examples of this include quantum dots[2–4] and graphene[5,6]. Downsizing inorganic semiconductors into quantum dots has led to enhanced optical, electrical, and magnetic properties because of quantum confinement effects[7], while nanosizing graphite into graphene leads to remarkable improvements in mechanical, electrical, and thermal properties[8].

Covalent organic frameworks (COFs)[9–12] have emerged as porous crystalline solids for applications such as gas adsorption, separation, and catalysis. Typically, COFs are used as micron-sized powdered aggregates that are formed under solvothermal synthesis conditions.

While such materials can be suitable for gas adsorption and related applications, the relatively large particle sizes (10's–100's mm) might limit effectiveness in optoelectronic process, particularly in photocatalysis. This is because light can only penetrate tens to hundreds of nanometers into conjugated organic photocatalysts because of their high molar extinction coefficients. Also, short photogenerated exciton diffusion lengths (~10 nm)[13] mean that internal excitons in bulk materials can be consumed by radiative or nonradiative exciton recombination before transport to the active catalyst surface[14]. While there are now numerous reports on COF photocatalysts[15–24], most studies have focused on chemical structure engineering to modify optoelectronic properties and wettability. Particle size and morphology have not so

[1]Leverhulme Research Centre for Functional Materials Design, Materials Innovation Factory and Department of Chemistry, University of Liverpool, Liverpool, UK. [2]Key Laboratory for Advanced Materials and Institute of Fine Chemicals, School of Chemistry and Molecular Engineering, East China University of Science and Technology, Shanghai, China. [3]Albert Crewe Centre for Electron Microscopy, University of Liverpool, Liverpool L69 3GL, UK. [4]Department of Materials Science and Engineering, Southern University of Science and Technology, 518055 Shenzhen, China. ✉e-mail: zhangweiwei@ecust.edu.cn; aicooper@liverpool.ac.uk

far been a major focus, even though a number of strategies for size-controlled COF synthesis have been reported[25–28]. As such, scope for improving photophysical properties of nanosized COF photocatalysts is underexplored.

Here, we present the bottom-up synthesis of two nano-COF colloids that exhibit exceptionally high photocatalytic activity for sacrificial hydrogen production. These two nano-COFs, TFP-BpyD and TFP-BD, were prepared in water in the presence of hexadecyl-trimethylammonium bromide (CTAB) and sodium dodecyl sulfate (SDS) surfactants. They have nanoribbon and nanosheet morphologies, respectively. In contrast to typical COF syntheses that involve crystallization under solvothermal conditions and require high vacuum to prevent oxidation of monomers, the synthesis of the nano-COFs here is facile. The process is conducted under ambient conditions using ultrasonication. Hence, this method is easily applicable for the bulk, scalable preparation of nano-COFs. Compared to their bulk COF analogues, these nano-COFs show improved particle dispersion in water and enhanced light-harvesting, leading to dramatically enhanced photocatalytic performance. In particular, the optimized TFP-BpyD nano-COF reached a hydrogen evolution rate of 392.0 mmol g$^{-1}$ h$^{-1}$, which to our knowledge is one of the highest mass-normalized sacrificial photocatalytic hydrogen evolution rates reported for organic materials so far. More importantly in terms of mechanistic understanding, we found a reverse concentration-dependent photocatalytic phenomenon in a COF system whereby adding *less* catalyst leads to *more* hydrogen production in absolute terms. To explain this, we carried out photoluminescence (FLS) and transient absorption spectra (TAS) measurements, which indicated anabatic singlet-singlet annihilation and charge-carrier recombination at higher nanoparticle concentrations due to increased interparticle collisions.

## Results

### Nano-COF synthesis and characterization

TFP-BpyD and TFP-BD nano-COF were selected for this study because of their high stability, irreversible enol-to-keto tautomerization, and good photocatalytic activity[29]. The synthesis procedures for TFP-BpyD and TFP-BD nano-COF were based on a previous report (Fig. 1a)[28]. The amine and aldehyde monomers were first loaded into CTAB/SDS mixed micelles to form separate homogeneous micellar solutions. By mixing these amine and aldehyde solutions with an acetic acid catalyst, the reaction mixture turned orange, indicating an imine condensation reaction. Unlike solvothermal syntheses and sonochemical syntheses[30], the reaction mixture remained clear and homogeneous with no apparent precipitation. A pronounced Willis-Tyndall scattering behavior from the two reaction mixtures confirms the presence of COF colloidal particles (Supplementary Fig. 1). To further verify the formation of nano-COF colloids, UV-visible absorption spectra and solution nuclear magnetic resonance (NMR) spectra were obtained directly for the colloidal solutions after 3 days. The $^1$H NMR spectra of TFP-BpyD and TFP-BD nano-COF revealed that the peak ($\delta \approx 9.8$ ppm) belonging to CHO hydrogens and the peaks ($\delta \approx 5.0$–$8.0$ ppm) attributed to NH$_2$ and aromatic hydrogens disappeared after the reaction (Supplementary Figs. 2, 3). UV-visible spectra showed that the characteristic peaks corresponding to the TFP, BpyD and BD monomers disappeared too, and new peaks were formed at 415 and 428 nm for TFP-BpyD and

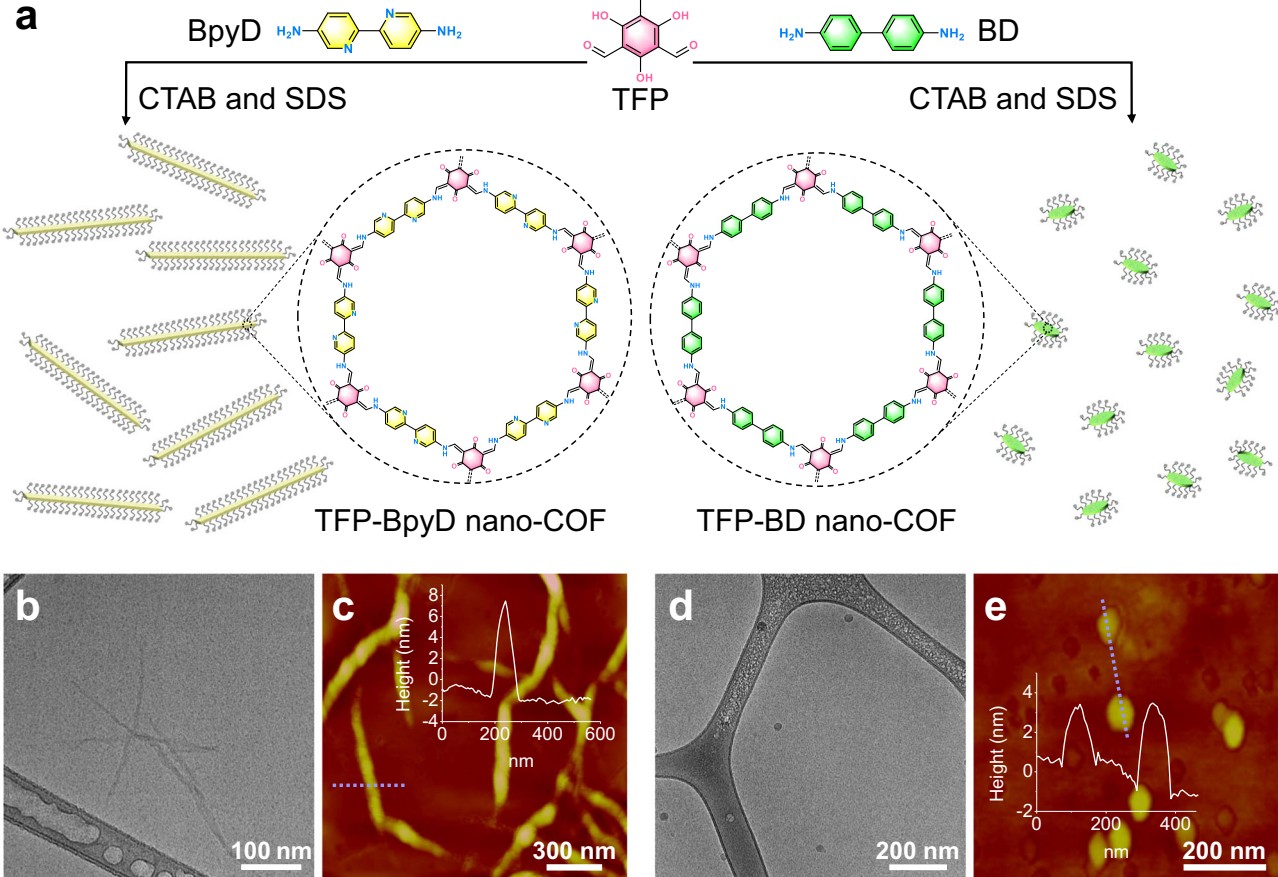

**Fig. 1 | Synthesis of nano-COFs and their morphologies.** The synthetic routes for TFP-BpyD and TFP-BD nano-COF (**a**). Cryo-TEM and AFM images of TFP-BpyD nano-COF (**b**, **c**) and TFP-BD nano-COF (**d**, **e**). Inserts of c and e: AFM height images with the corresponding cross-section analysis along the indicated dashed blue lines. Source data for (**c**) and (**e**) are provided as a Source Data file.

TFP-BD nano-COF, respectively, further proving that the starting materials were consumed, and that condensed, polymeric imine species were formed (Supplementary Fig. 4).

To further characterize the nano-COFs, ammonia and ethanol were added to the reaction mixtures, which precipitated the materials as insoluble yellow powders. Elemental analysis (EA) for the solid nano-COFs corroborates their structure (Supplementary Table 1). Fourier-transform infrared (FT-IR) spectra revealed the appearance of stretching bands at around 1570 cm$^{-1}$ and 1286 cm$^{-1}$, which are assigned to the C = C and CH−NH bonds (Supplementary Fig. 5). $^{13}$C CP-MAS solid-state NMR spectra showed clear signals near 184 and 148 ppm, corresponding to the carbonyl carbons and −CH-NH− (Supplementary Fig. 6), respectively. The full X-ray photoelectron spectroscopy (XPS) survey spectra confirmed the presence of C, N, and O elements of both nano-COFs (Supplementary Fig. 7a). High-resolution XPS spectra of C 1$s$ for both nano-COFs show the presence of ketoenamine C = O (Supplementary Fig. 7b). All above results indicate the formation of β-ketoenamine linkage in the two nano-COFs. High-resolution XPS spectra of N 1$s$ for TFP-BpyD nano-COF shows two individual peaks at 399.1 and 397.9 eV, attributable to the keto-enamine N (−C − HN − C−) and pyridinic N (−C = N − C−), respectively (Supplementary Fig. 7c), where the more electronegative pyridinic N would be favorable for photocatalysis. Powder X-ray diffraction (PXRD) patterns of isolated nano-COFs showed no obvious peak from the monomers, further supporting the complete consumption of the amine and aldehyde monomers (Supplementary Fig. 8). PXRD suggested a slightly higher degree of structural order in TFP-BpyD nano-COF than TFP-BD nano-COF, possibly due to higher reversibility in the BpyD linker, although neither COF showed much evidence of long-range order and crystallinity in this precipitated, bulk form. A degree of permanent porosity was confirmed by nitrogen adsorption-desorption experiments at 77 K for these bulk precipitated powders (Supplementary Figs. 9, 10). Both nano-COFs exhibited a rapid uptake at low pressure $P/P_o$ < 0.1 with a sorption profile best described as a type IV isotherm, characteristic of microporous / mesoporous material. The Brunauer-Emmett-Teller (BET) surface areas of TFP-BpyD and TFP-BD nano-COF were calculated to be 113 m$^2$ g$^{-1}$ and 598 m$^2$ g$^{-1}$, respectively. Here, both nano-COFs showed lower crystallinity and porosity compared with bulk synthesized COFs. This stems from the lower reversibility of the covalent bond formation that is associated with enol-to-keto tautomerization, as well as the reduced crystallite sizes, and reduced long-range order[29]. Thermogravimetric analysis (TGA) indicated that these nano-COFs were thermally stable up under nitrogen to around 350 °C (Supplementary Fig. 11).

The morphology of the two nano-COFs was characterized using SEM and TEM. These images suggested that TFP-BpyD and TFP-BD nano-COF had nanofiber and nanosphere morphologies, respectively (Supplementary Figs. 12–15). However, aggregation in the precipitated powders made it difficult to discern the real size of the primary particles for the two nano-COFs. We therefore used cryo-TEM to observe the morphology of the two nano-COFs closer to their native state in solution. TFP-BpyD nano-COF showed a nanofiber morphology with fiber lengths around 300 nm and fiber diameters of 10–20 nm (Fig. 1b and Supplementary Fig. 16a). Cryo-TEM suggested that TFP-BD nano-COF had a 'spherical' morphology with a particle diameter of around 30 nm (Fig. 1d and Supplementary Fig. 16b). AFM images confirmed this particle size, but the average heights of the two nano-COFs as measured by AFM were 8 nm and 4 nm, respectively, suggesting that nanoribbons (i.e., flat fibers) and nanosheets (i.e., disks) might be a more accurate description of the two particle morphologies (Fig. 1c, e and Supplementary Fig. 17). This morphology might stem from the strong tendency of these COFs to formed two-dimensional layered packings, and can explain the low crystallinity due to the lack of longer-range π−π stacking. The reason for the difference in morphologies between the two nano-COFs, which have similar chemical structures (Fig. 1a), is unclear.

## Photocatalytic hydrogen evolution reaction

To study the effect of particle size on photocatalytic activity, the bulk COF analogues of these two nano-COFs, TFP-BpyD COF and TFP-BD COF, were synthesized using our sonochemical method[30] as a control. These bulk COFs were characterized by EA (Supplementary Table 1), FT-IR spectroscopy, SEM, nitrogen adsorption-desorption measurements, UV-visible absorption spectroscopy, PXRD and TGA measurements (see Supplementary Information). The bulk materials showed FT-IR spectra that were similar to the precipitated TFP-BpyD and TFP-BD nano-COFs as well as similar thermal stabilities (Supplementary Figs. 11 and 18), but the bulk materials showed much higher long-range order (c.f., Supplementary Figs. 19, 20) and, as a result, significantly higher surface areas (1149 m$^2$ g$^{-1}$ and 630 m$^2$ g$^{-1}$, respectively, Supplementary Figs. 21, 22). SEM images showed micron-scale aggregates for the bulk COFs (Supplementary Figs. 23, 24). Notably, when dispersing nano-COFs and bulk COF materials at the same concentrations in aqueous solution, the nano-COFs showed far better dispersion and more effective light-harvesting than the comparable bulk COFs. As shown in Fig. 2a and Supplementary Fig. 25, both nano-COFs show low light transmittances (10–25 %) in the wavelength range 350–550 nm, while the corresponding bulk COF dispersions absorb much less light in this range (70–90% transmission); that is, the two nano-COFs are far more effective at light harvesting at these concentrations. Compared to bulk COFs, the maximum absorption band of the nano-COFs showed a blue shift of around 100 nm. This can be attributed to the decreased π-conjugation in the $ab$ plane, as well as reduced stacking in $c$-axis.

Photocatalytic hydrogen production experiments were carried out with both nano-COFs and bulk COFs as the photocatalysts, using ascorbic acid (AA) as the sacrificial agent and platinum as a cocatalyst. Here, we selected AA as a sacrificial electron donor because of its strong reducibility for ketoenamine COFs, as evidenced by previous work[18]. As shown in Fig. 2b, TFP-BpyD bulk COF powder (5 mg) produced 23.2 μmol H$_2$ in 5 h under visible light, with a mass-normalized hydrogen evolution rate (HER) of 0.89 mmol g$^{-1}$ h$^{-1}$ (4.5 μmol h$^{-1}$). For the corresponding nano-COF, the same photocatalytic experiment was carried out by diluting a stock solution of the nano-COF colloid with water. Remarkably, by using 0.25 mL of the TFP-BpyD nano-COF stock colloid solution, which contains just 0.085 mg of the COF, we generated 162.3 μmol H$_2$ in 5 h. The amount of H$_2$ evolved for the TFP-BpyD nano-COF was 7-fold higher than for the bulk TFP-BpyD material in absolute terms, while the enhancement in the mass-normalized HER was a factor of 440, achieving an average HER of 392.0 mmol g$^{-1}$ h$^{-1}$ (33.3 μmol h$^{-1}$). Similarly, the TFP-BD nano-COF showed enhanced photocatalytic H$_2$ production compared with bulk TFP-BD COF (Supplementary Fig. 26) with an average HER of 183.0 mmol h$^{-1}$ g$^{-1}$ (15.6 μmol h$^{-1}$). Even though the bulk COFs showed improved crystallinity, higher levels of porosity, and a larger thermodynamic driving force for proton reduction (Supplementary Fig. 35), all of which are important factors for photocatalytic efficiency, the nano-COFs exhibited greatly enhanced hydrogen evolution because of their much better light harvesting characteristics.

We sought next to study the effect of the colloid concentration on the photocatalytic performance of the TFP-BpyD nano-COF. As shown in Fig. 2c, decreasing the volume of TFP-BpyD nano-COF added from 1 to 0.5, 0.1 and 0.05 mL (corresponding to nano-COF concentrations of 68.6, 34.3, 6.86 and 3.43 μg/ mL, respectively) led to a roughly 10-fold increase in the absolute amount of hydrogen produced, form 0.4 mmol to 3.5 mmol. This differs markedly from previous studies involving bulk COF and polymer catalysts[31,32], where the amount hydrogen evolved barely changes when increasing the concentration of photocatalyst up to some saturation value. We discuss this

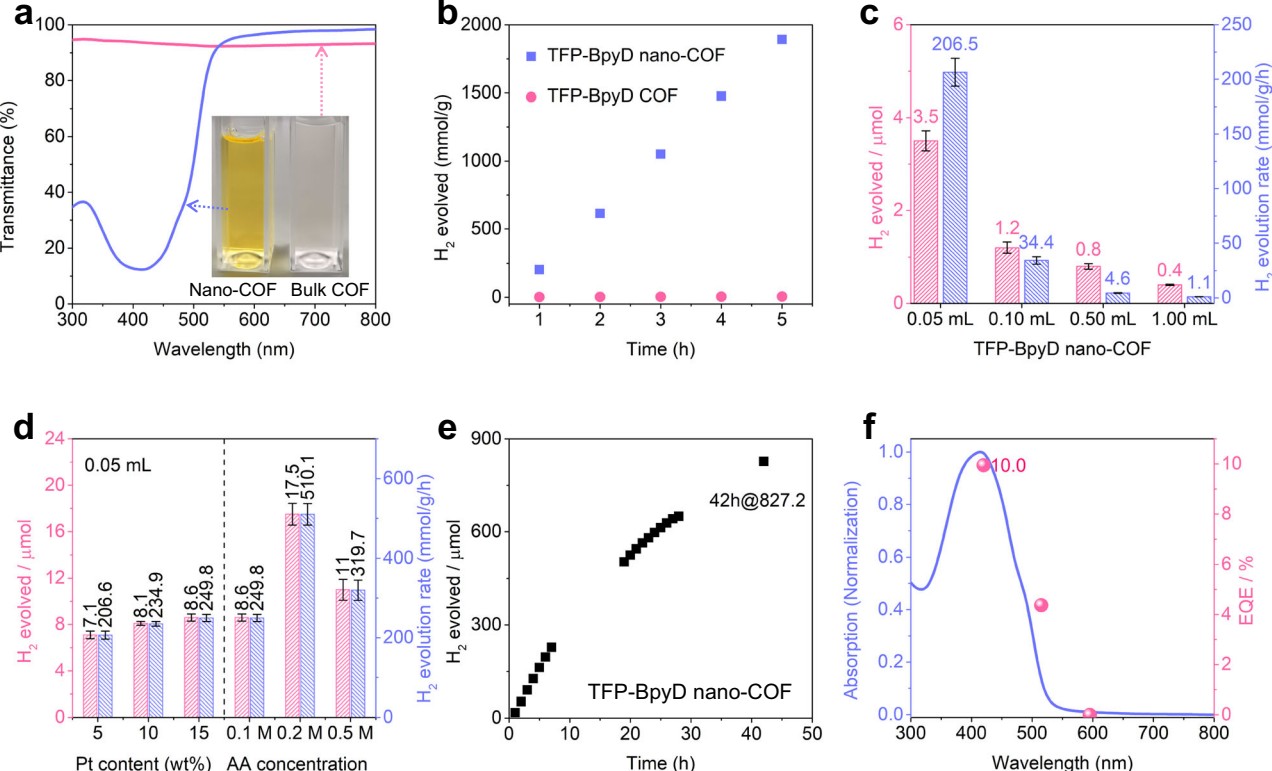

**Fig. 2 | Photocatalytic hydrogen evolution for bulk COFs and nano-COFs, and inverse concentration dependence for TFP-BpyD nano-COF. a** Transmittance spectra of TFP-BpyD nano-COF (blue) with bulk TFP-BpyD COF (pink) at the same concentration (11.4 μg/mL); see also Supplementary Fig. 25. **b** Comparison of photocatalytic $H_2$ evolution of TFP-BpyD nano-COF (blue squares) with bulk TFP-BpyD COF (pink circles) over 5 h (conditions for nano-COF: 0.25 mL TFP-BpyD nano-COF in 24.75 water, diluted $H_2PtCl_6$ solution as a platinum precursor (15 wt.% loading), 880 mg AA (overall concentration is 0.2 M), λ > 420 nm; Conditions for bulk COFs: 5 mg of bulk COF catalyst in water, diluted $H_2PtCl_6$ solution as a platinum precursor (4 wt.% loading), 0.2 M AA (25 mL), λ > 420 nm). **c** Sacrificial photocatalytic $H_2$ evolution using different concentrations of TFP-BpyD nano-COF. Conditions were as follows: TFP-BpyD nano-COF (1.0, 0.5, 0.1 and 0.05 mL) in 0.1 M AA solution (5 mL in final), 5 wt.% Pt loading, 1 h illumination (Oriel Solar Simulator,

1.0 sun). **d** Effect of varying Pt and AA concentration for TFP-BpyD nano-COF. For Pt concentration series, TFP-BpyD nano-COF (0.05 mL) in 0.1 M AA solution (5 mL total volume), 5, 10, or 15 wt.% Pt loading, 2 h illumination; For AA concentration series, TFP-BpyD nano-COF (0.05 mL) in 0.1, 0.2 or 0.5 M AA solution (5 mL total volume), 15 wt.% Pt loading, 2 h illumination. Blue bars, $H_2$ evolution rate; pink bars, $H_2$ evolution amount. **e** Plot showing sacrificial photocatalytic $H_2$ evolution as a function of time for TFP-BpyD nano-COF over 42 h (0.25 mL TFP-BpyD nano-COF in 24.75 water, diluted $H_2PtCl_6$ solution as a platinum precursor (15 wt.% loading), 880 mg AA (overall concentration is 0.2 M), λ > 420 nm. **f** Overlay of the UV/vis absorption spectrum (blue curves) of TFP-BpyD nano-COF with external quantum efficiency (EQE, pink dots) at three different incident light wavelengths. Source data for Fig. 2 are provided as a Source Data file.

counterintuitive 'reverse-concentration' phenomenon below. We also optimized the amount of Pt and the AA sacrificial electron donor (Fig. 2d); the best condition was found to be 0.00343 mg/ mL TFP-BpyD nano-COF (that is, the original colloidal solution diluted 100 times) with 0.2 M AA and 15 wt. % Pt loading with respect to the COF. The longer-term photocatalytic performance of this optimized TFP-BpyD nano-COF system was investigated. As shown in Fig. 2e and Supplementary Fig. 36, TFP-BpyD nano-COF showed sustained $H_2$ productions up to 42 h in the presence of AA as a sacrificial electron donor and Pt as a cocatalyst, albeit with a decrease in rate after around 20 h, giving an average HER of 392.0 mmol $h^{-1}$ $g^{-1}$ over the first 5 h under visible light (λ > 420 nm, 300 W Xenon lamp). To our knowledge, this is one of the highest mass-normalized sacrificial photocatalytic $H_2$ evolution rates reported for an organic photocatalyst so far (Supplementary Table 3). The external quantum efficiency (EQE) was measured at different wavelengths to evaluate the photocatalytic $H_2$ production performance. The EQE was determined to be 10.0% and 5.5% at 420 nm for TFP-BpyD and TFP-BD nano-COF, respectively, and the EQE followed the absorption spectrum, supporting a photo-induced $H_2$ evolution process (Fig. 2f and Supplementary Fig. 37). After photocatalysis, the TFP-BpyD and TFP-BD nano-COF materials were characterized by TEM, STEM, SEM, NMR and FT-IR. TEM image showed the retention of nanoscale morphology and Pt nanoparticle were

observed to be distributed uniformly on the nano-COFs (Supplementary Figs. 38, 39). HAADF-STEM images and elemental mappings (Supplementary Figs. 40, 41) also indicated good Pt cocatalyst dispersion on the nano-COFs. FTIR and NMR characterizations indicated no significant changes after photocatalysis (Supplementary Figs. 42, 43), while TEM and SEM (Supplementary Fig. 44) analysis suggested some aggregation of the COF crystallites. We therefore consider the aggregation of nano-COFs to be the main reason for nonlinearity of the photocatalytic activity after 20 h, rather than the destruction of the COF skeleton. The depletion of AA and/or the accumulation of AA degradation products may also hinder the HER, as discussed in previous studies[33,34].

## Mechanistic study

To better understand the reverse concentration dependence for $H_2$ evolution for the TFP-BpyD nano-COF (Fig. 2c), we measured UV-vis absorption, photoluminescence spectra (PL) and transient absorption spectra (TAS) for nano-COF aqueous solutions at different colloid concentrations. The UV-vis absorption spectra showed a concentration-dependent absorption for nano-COFs, and the position and shape of the absorption band were unchanged at these different concentrations (Supplementary Figs. 45–48), suggesting that the nano-COF was in a dispersed state without any apparent particle

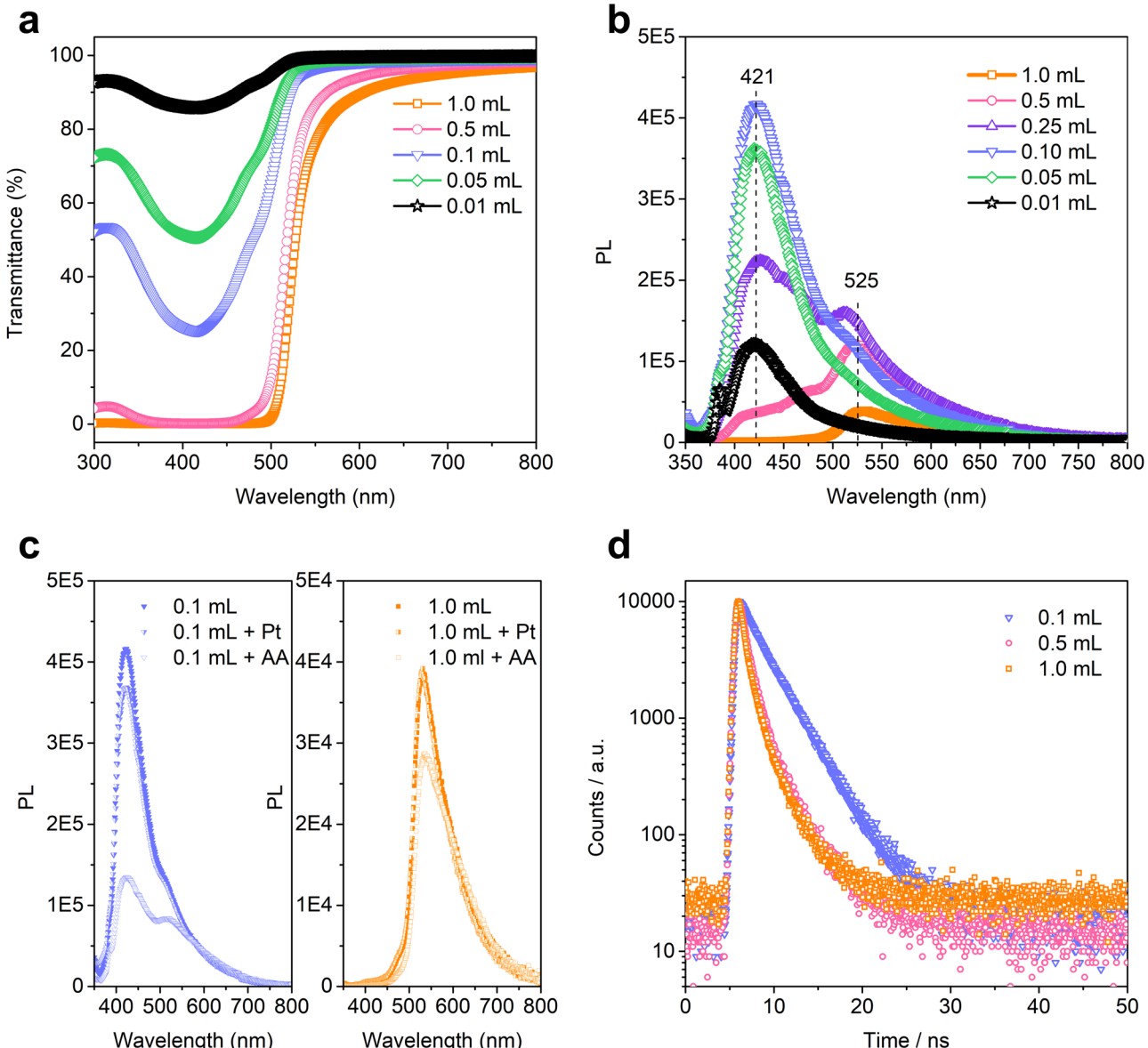

**Fig. 3 | Photophysical properties of TFP-BpyD nano-COF solutions at various concentrations. a** Transmittance spectra of TFP-BpyD nano-COF at different concentrations (conditions: 0.01, 0.05, 0.10, 0.50 and 1.00 mL of the stock TFP-BpyD nano-COF solutions were diluted with water; the total solution volume is 5 mL). **b** Photoluminescence (PL) spectra of TFP-BpyD nano-COF with different concentrations (conditions: 0.01, 0.05, 0.10, 0.25, 0.50 and 1.00 mL of the stock TFP-BpyD nano-COF solutions were diluted with water; the total solution volume is 5 mL). **c** PL spectra recorded in the presence of AA (0.10 and 1.00 mL of the TFP-BpyD nano-COF stock solutions were diluted by 0.2 M AA, open symbols) and Pt (0.10 and 1.00 mL of the TFP-BpyD nano-COF stock solutions were diluted by water with Pt co-catalyst at 15 wt. % loading, half-closed symbols). The total solution volume is 5 mL. **d** TCSPC experiment for TFP-BpyD nano-COF at different concentrations in water (5 mL); the volumes refer to the amount of stock colloid solution added. 0.01, 0.05, 0.10, 0.25, 0.50 and 1.00 mL are represented by black stars, green diamonds, blue lower triangles, purple upper triangles, pink circles and yellow squares, respectively. Source data for Fig. 3 are provided as a Source Data file.

aggregation over the concentration ranges studied here. Transmittance spectra of TFP-BpyD nano-COF indicated increased light-harvesting when increasing the amount of nano-COF solution added from 0.01 to 1 mL (in a 5 mL total volume). When 0.5 mL and 1.0 mL of these stock TFP-BpyD nano-COF solution was added, the light absorption was saturated in the range of 350 to 550 nm, indicating total light absorbance (Fig. 3a).

When we used less than 0.1 mL of TFP-BpyD nano-COF of the stock solution (5 mL total volume), an intense emission peak was observed at 421 nm upon excitation at 340 nm (Fig. 3b). The intensity of this emission peak showed a maximum at 0.1 mL. Increasing the concentration of the nano-COF beyond 0.1 mL to 0.25 mL or higher

(Fig. 3b) caused significant PL quenching and a dramatic reduction in the intensity of this emission peak. Also, at higher TFP-BpyD nano-COF concentrations (0.25 mL), a dual-emission was observed with an additional emission band emerging at 525 nm (E1 @ 421 nm, E2 @ 525 nm; Fig. 3b). The different lifetimes of E1 and E2 (Supplementary Fig. 49) and the redshift of the excitation spectra (Supplementary Fig. 50) indicated two different dynamic processes. We assign E1 to singlet emission from an individual nano-COF particle based on the small Stokes-shift, while E2 with a larger Stokes-shift can be assigned to a radiative decay process due to the charge-transfer (CT) that occurs between the nano-COF particles at higher concentrations. We believe that this PL change as a function of nano-COF concentration

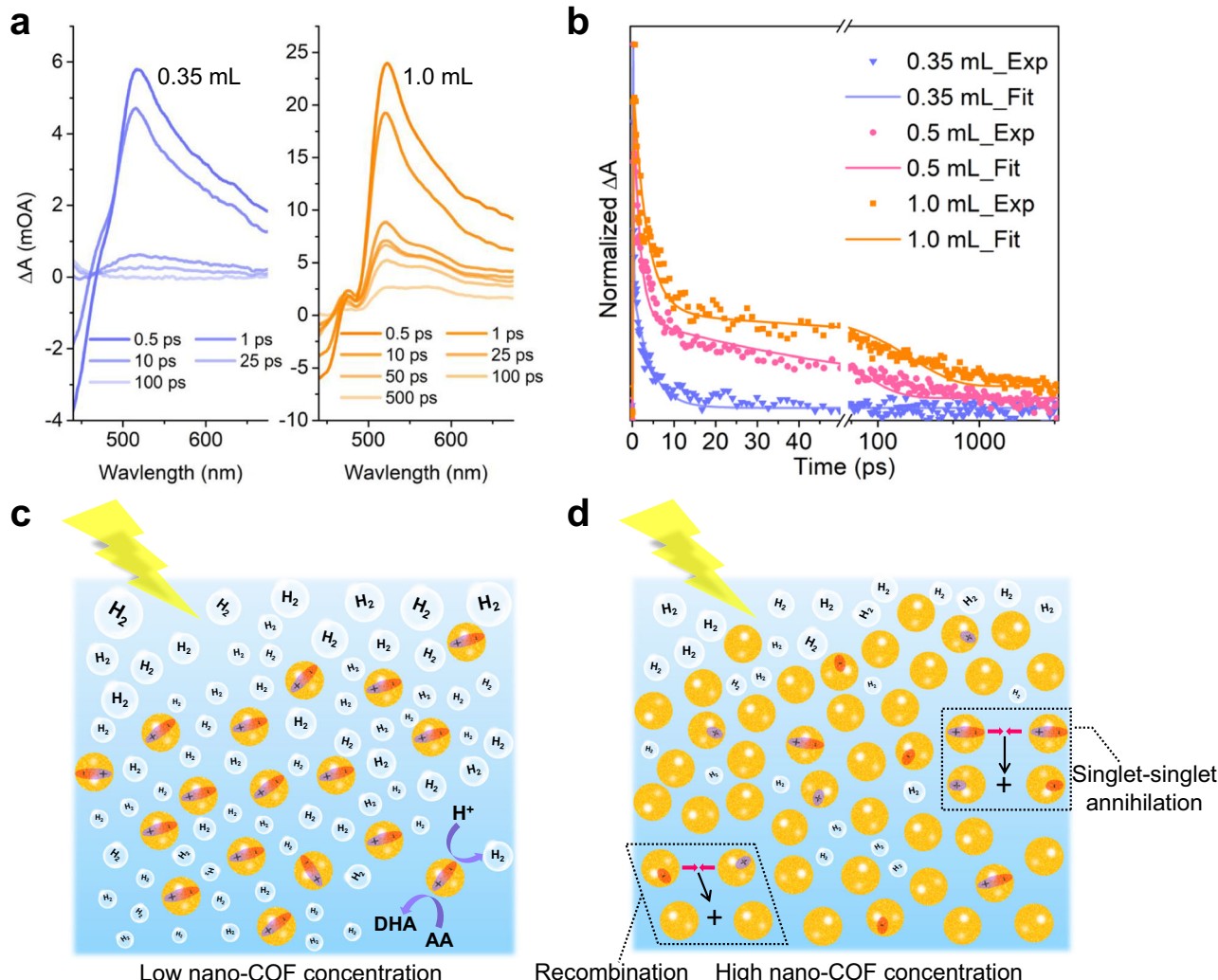

**Fig. 4 | Transient absorption spectra (TAS) for TFP-BpyD nano-COF at different concentrations and mechanistic interpretation. a** TAS spectra under the following conditions: 0.35 mL (blue) or 1.0 mL (yellow) TFP-BpyD nano-COF, diluted by water to a total volume is 5 mL. **b** Life time test of TFP-BpyD nano-COF by TAS measurement. 0.35, 0.50 and 1.00 mL are represented by the color of blue, pink and yellow, respectively. The experimental and fitted data correspond to the symbols and curves. Scheme showing free charge carriers produced for nano-COFs at (**c**) low concentration and (**d**) high concentration (ascorbic acid: AA; dehydroascorbic acid: DHA). Source data for Fig. 4a and b are provided as a Source Data file.

is analogous to the small molecule pyrene, which forms an excimer with a bathochromic shift in its emission peak and, likewise, shows emission quenching at higher concentrations (Supplementary Fig. 51)[35–37]. Typically, the formation of excimer in small molecules is promoted by monomer density (here the concentration of nano-COF), because of the dependence on a bimolecular interaction. Also, excimers usually show longer emission wavelengths than the excited monomer's emission (here the emission of a single nano-COF particle)[35]. As such, it seems that this nano-COF exhibits a molecule-like excitonic nature, showing CT character at high particle concentrations.

Both emission bands of E1 and E2 were decreased when further increasing the concentration of TFP-BpyD nano-COF to 0.5 and 1 mL (Fig. 3b). We attribute this to the increased collision rate of the nano-COF particles at higher concentrations, which suppresses the radiative recombination of free charge carriers. The magnitude of the PL quench for E1 was more pronounced than E2, which bears charge-transfer character, indicating that particle collision has a greater impact on the singlet emission (E1). Adding a sacrificial donor (AA) to the TFP-BpyD nano-COF solution gave rise to a similar quenching phenomenon (Fig. 3c and Supplementary Fig. 52). Again, the

magnitude for the PL quench with AA for E1 greater than for E2, suggesting that singlet emission is more easily be extracted by AA. TSCPC was also carried out with TFP-BpyD nano-COF for emission at 371 nm. The lifetime for TFP-BpyD nano-COF with low-concentration (0.1 mL, $\tau_{avg}$ = 3.04 ns) was estimated to be significantly longer than at higher nano-COF concentrations (0.5 mL, $\tau_{avg}$ = 1.39 ns and 1.0 mL, $\tau_{avg}$ = 1.18 ns, Fig. 3d and Supplementary Figs. 53–55), supporting an interparticle charge transfer process at high nano-COF concentrations. We note that the corresponding bulk materials did not show an equivalent PL quench phenomenon at higher COF concentrations (Supplementary Figs. 57, 58).

Femtosecond (fs) transient absorption (TA) measurements were performed to study the dynamic processes for the TFP-BpyD nano-COF (Fig. 4a and Supplementary Fig. 64). The TFP-BpyD nano-COF exhibited a broadband photoinduced absorption (PIA) signal around 510 nm, independent of the concentration, which could be attributed to polaron pairs originating from individual nano-COF parieles[38,39]. At lower nano-COF concentration (0.35 mL), this PIA signal was found to decay very rapidly. By comparison, at higher nano-COF concentrations, a long-lifetime process was also observed for the PIA signals (Fig. 4b). This long lifetime may result from increased particle collision

rates at high concentrations that induce singlet-singlet annihilation, note that the polaron pairs and separated charges can give rise to similar spectral signatures[40,41].

The reverse concentration-dependent photocatalytic performance can be rationalized by these various observations. At low nano-COF concentration, a greater number of electron-hole pairs (the initial excitation, E1) can be generated and extracted by the sacrificial agent, AA, to promote photocatalytic proton reduction. However, increasing the COF concentration causes increased collision rates between the nano-COF particles, together with a charge-transfer process and singlet-singlet annihilation, which finally produces free charges that are spatially isolated, along with longer lifetimes than at low concentrations (Fig. 4c, d)[38]. Given that this charge transfer state is barely extracted by AA, we ascribe this loss of the initial excitation state and resulting anabatic charge recombination to be the primary cause for the decreased sacrificial photocatalytic activity. In addition to these effects, at even higher concentrations (>1 mL nano-COF stock solution), the solutions become totally absorbing (Fig. 3a); as such, one would expect the mass-normalized hydrogen evolution rate to decrease in this regime, irrespective of interparticle charge transfer effects.

## Discussion
Two nano-COFs, TFP-BpyD and TFP-BD, were synthesized in water by using surfactants. These nano-COFs showed greatly improved dispersion in water and much more effective light-harvesting as compared to their bulk counterparts. This resulted in a 440-fold enhancement in the sacrificial photocatalytic hydrogen evolution rate. In particular, the TFP-BpyD nano-COF showed an unprecedented mass-normalized $H_2$ evolution rate up to 392.0 mmol $g^{-1}$ $h^{-1}$, which is one of the highest rates reported so far for an organic photocatalyst. A reverse concentration-dependent photocatalytic performance was found for these nano-COFs, which has not been observed before and is not a property of the corresponding bulk COF materials. Photophysical measurements suggest that collision between nano-COF particles at higher concentrations lead to charge-transfer processes that result in singlet-singlet annihilation, limiting the generation of active species for photocatalysis. In addition to highlighting the fundamental importance of particle size for COF photocatalysts, these finding suggest several future directions. For example, such nano-COFs might find uses as earth-abundant replacements for precious-metal photocatalysts for organic transformations. Moreover, the high catalytic activity of these materials coupled with their high surface-to-volume ratio suggests the possibility of using these materials as components in solution phase or solid-state Z-schemes for overall water splitting[42] or for $CO_2$ reduction[43].

## Methods
1,3,5-Triformylphloroglucinol (TFP), benzidine (BD) and acetic acid were obtained from Fluorochem Ltd. [2,2′-Bipyridine]-5,5′-diamine (BpyD) was obtained from BLD pharm Ltd. Hexadecyl trimethyl ammonium bromide (CTAB) and sodium dodecyl sulfate (SDS) were purchased Sigma-Aldrich Ltd. Anhydrous solvents were purchased from Acros Organics or Fisher Scientific. All chemicals were used as received without further purification.

### Synthesis of TFP-BpyD nano-COF in water
TFP (21 mg, 0.1 mmol) was dissolved in 1 mL of dimethyl sulfoxide (DMSO). This TFP solution was added dropwise to a flask containing 58 mL of 0.05 M CTAB aqueous solution. After ultrasonication, 1.8 mL of 0.05 M SDS aqueous solution was added to form solution A. Separately, BpyD (28 mg, 0.15 mmol) of was dissolved in 0.5 mL of DMSO. The solution was added dropwise to a flask containing 58 mL of 0.05 M CTAB aqueous solution. After ultrasonication, 1.8 mL of 0.05 M SDS solution was added to form solution B. Finally, solutions A and B were

mixed, and 5.8 mL of acetic acid was added. After reacting at room temperature for 72 h, an orange colloidal solution was formed. The ideal concentration of TFP-BpyD nano-COF was 0.343 mg mL$^{-1}$, assuming complete monomer conversion.

### Synthesis of TFP-BD nano-COF in water
As for the synthesis procedure for TFP-BpyD nano-COF, 10.5 mg (0.05 mmol) TFP and 13.8 mg (0.075 mmol) of BD were used to form solution A and solution B. The ideal concentration of TFP-BD nano-COF was 0.170 mg mL$^{-1}$, assuming complete monomer conversion.

### Isolation of nano-COFs to obtain bulk solids
The TFP-BpyD nano-COF and TFP-BD nano-COF colloids were neutralized with 6.8 mL of concentrated ammonia and 100 mL of ethanol, yielding a yellow solid precipitate. The dispersion was centrifuged for 5 min at 1118 × $g$, and the supernatant was removed. The solid was redispersed in 50 mL of ethanol and stirred for 30 min before centrifuging again. This washing procedure was repeated a total of 5 times, and a highly concentrated dispersion in ethanol was obtained. Finally, the sample was activated by supercritical $CO_2$ drying to obtain bulk TFP-BpyD nano-COF (yield: 61%) and TFP-BD nano-COF (yield: 69.2%) powders.

### Synthesis of bulk COFs
The TFP and BpyD or BD monomers and 12 M acetic acid were added into a vial and sonicated (550 W Branson Sonifier SFX550 cell disrupter with O.D. 3 mm microtip) in continuous mode for 60 min. The details could be found in the Supplementary Table 2. The resulting solid was washed in sequence with acetone and methanol, followed by a 24 h Soxhlet extraction using THF. The sample was then washed and immersed in hexane (12 h) to replace all other solvents and dried under high vacuum at 80 °C for 24 h to obtain bulk TFP-BpyD COF (yield: 93%) and TFP-BD COF (yield: 66%).

### Solution nuclear magnetic resonance (NMR)
$^1$H spectra were recorded in solution on a Bruker Avance 400 NMR spectrometer at 400 MHz and 100 MHz, respectively.

### Fourier-transform infrared spectroscopy (FT-IR)
IR spectra were recorded on a Bruker Tensor 27 FT-IR spectrometer using KBr pellets.

### Powder X-ray diffraction (PXRD)
PXRD measurements were performed on a PANalytical X'Pert PRO MPD, using in high throughput transmission mode with Kα focusing mirror and PIXCEL 1D detector with a Cu X-ray source.

### Structural modeling of COFs
Structural atomistic simulations of the possible framework structures were carried out using *Material Studio* software. The simulated PXRD patterns were determined by the Reflex module. The space group for simulated structures was selected as P1.

### Thermogravimetric analysis (TGA)
TGA analysis was performed on an EXSTAR6000 under a flow of nitrogen by heating from room temperature to 800 °C at a rate of 10 °C min$^{-1}$.

### Gas sorption analysis
The surface areas and nitrogen adsorption isotherms of samples (at 77.3 K) were obtained using a Micromeritics ASAP 2020 volumetric adsorption analyzer. Before analysis, the samples were degassed at 120 °C for 12 h under vacuum ($10^{-5}$ bar). Pore size distributions of COFs were obtained by fitting the density functional theory (DFT) model to the adsorption data.

### Elemental analysis (EA)

EA was determined on Thermo EA1112 Flash CHNS-O analyzer with standard microanalytical procedures.

### X-ray photoelectron spectroscopy (XPS)

XPS data were measured in powder (without surfactants) form using an ESCALAB 250Xi instrument (Thermo Fisher Scientific) with a monochromatized Al Kα line source.

### Solid-state $^{13}$C CP (Cross Polarization) MAS (Magic Angle Spinning) NMR

Solid-state $^{13}$C NMR experiments were performed under 25 °C on a Bruker Avance III HD 400WB (9.4 T) spectrometer, operating at 100.6 MHz for $^{13}$C, at a spinning speed of 13 kHz on a 4.0 mm CPMAS probe. Activated samples were packed into zirconia rotors inside a glove box under nitrogen. CP spectra were acquired under dry air with $^{1}$H p/2 pulse of 2.3 ms, a contact time of 3 ms and a recycle time of 2 s.

### Scanning electron microscopy (SEM)

SEM images were collected on a Hitachi S-4800 cold field emission scanning electron microscope. Samples were prepared by depositing the dry powders on 15 mm Hitachi M4 aluminum stubs using an adhesive high-purity carbon tape before coating with a 2 nm layer of gold using an Emitech K550X automated sputter coater.

### Transmission electron microscopy (TEM)

TEM images were obtained on a JEOL 2100FCs microscopy at an accelerating voltage of 200 kV. The samples were prepared by drop-casting sonicated ethanol suspensions of the materials onto a copper grid.

### Cryo-Transmission Electron Microscopy (Cryo-TEM)

The Cryo-TEM was carried out with a FEI Talos F200X G2 and this microscope was optimized for liquid samples. Specimen preparation was carried out as follow: a small drop of FNP sample was deposited on lacey carbon coated-300 mesh copper grid. Excess liquid was removed by blotting with a filter paper, leaving a thin film of the solution on the grid. The sample was vitrified in liquid ethane and transferred to the microscope, continuously kept below −170 °C and protected against atmospheric conditions. The analysis was performed by setting the microscope at an acceleration voltage of 200 kV and the high-resolution images were recorded under low electron conditions with a camera of Ceta 4 K*4 K 200 kV."

### Scanning transmission electron microscopy (STEM)

STEM and element mapping were collected on a Talos F200X transmission electron microscope, and samples were casted on a copper grid with holey carbon supporting films.

### Transient absorption spectroscopy (TAS)

The femtosecond pump-probe TAS measurements were performed using a regenerative amplified Ti:sapphire laser system (Coherent; 800 nm, 70 fs, 6 mJ/pulse and 1 kHz repetition rate) as the laser source and a Femto-100 spectrometer (Time-Tech LLC). Briefly, the 800 nm output pulse from the regenerative amplifier was split in two parts with a 50% beam splitter. The transmitted part was used to pump a TOPAS Optical Parametric Amplifier (OPA) which generated a wavelength-tunable laser pulse from 250 nm to 2.5 μm as pump beam. The reflected 800 nm beam was split again into two parts. One part with <10% was attenuated with a neutral-density filter and focused into a CaF2 crystal to generate a white light continuum (WLC) used for probe beam. The probe beam was focused with an Al parabolic reflector onto the sample. After the sample, the probe beam was collimated and then focused into a fiber-coupled spectrometer with CMOS sensors and detected at a frequency of 1 KHz. The intensity of the pump pulse used in the experiment was controlled by a variable neutral-density filter wheel. The intensity of the pump pulse was 400 μw and pump wavelength was 400 nm. The delay between the pump and probe pulses was controlled by a motorized delay stage. The pump pulses were chopped by a synchronized chopper at 500 Hz and the absorbance change was calculated with two adjacent probe pulses (pump-blocked and pump-unblocked). The samples were placed in 1 mm airtight cuvettes and were measured in an argon atmosphere.

### Atomic force microscopy (AFM)

AFM images were obtained on a Bruker Dimension ICON. The data were processed using NanoScope Analysis 2.0. Operation mode: Tapping mode in air. The radius of curvature of the tip: ~5 nm.

### UV-visible absorption spectra

UV-visible absorption spectra of the polymers were measured on a Shimadzu UV-2550 UV-vis spectrometer by measuring the reflectance of powders in the solid state.

### Cyclic voltammetry (CV)

CV curves were carried out using a Bio-logic SP200 electrochemical workstation in a normal three electrode cell (glassy carbon as the working electrode, Pt wire as counter electrode and Ag/AgCl electrode as the reference electrode). The experiments were carried out in acetonitrile solutions with 0.1 M tetra-$n$-butylammonium hexafluorophosphate (TBAPF$_6$) as the supporting electrolyte at a scan rate of 100 mV s$^{-1}$. The ferrocenium/ferrocene ($F_c/F_{c^+}$) redox couple was used as an external potential reference.

$$E_{HOMO} = -\left(E_{ox,onset} - \frac{E^{(\frac{1}{2})}F_c}{F_c{}^+} + 4.8\right) eV \text{ vs. vacuum} \qquad (1)$$

$$E_{LUMO} = E_{HOMO} + E_g \text{ vs. vacuum} \qquad (2)$$

$$\text{Reference} : \frac{E^{(\frac{1}{2})}F_c}{F_c{}^+} = \frac{0.10 + 0.03}{2} = 0.065 \qquad (3)$$

### High-throughput photocatalytic H$_2$ production experiment

Nano-COF colloids were added into sample vials (volume = 12.5 mL) and purged with nitrogen in a Chemspeed Technologies Sweigher robot for 6 h. The robotic liquid transfer head was used to transfer sacrificial electron donor aqueous solution and H$_2$PtCl$_6$ via the liquid handling system under theses inert conditions from stock jars inside the system into the sample vials. The capper/crimper tool then capped/crimped the vials automatically, again under inert conditions. All sample vials were ultrasonicated in an ultrasonic bath for 10 min before removing the vials and illuminating them with a solar simulator (AM1.5 G, Class AAA, IEC/JIS/ASTM, 1440 W Xe, 12 × 12 in., MODEL:94123A) for 1 h while constantly being agitated with a rocker/roller device. Gaseous products were analyzed on an Agilent HS-GC injecting a sample from the headspace via a transfer line (temperature 90 °C) onto a 5 Å molecular sieve column (temperature: 45 °C) with helium as the carrier gas at a flow rate of 30 mL min$^{-1}$. Hydrogen was detected with a pulsed discharge detector referencing against standard gas with a known concentration of hydrogen. Any hydrogen dissolved in the reaction mixture was not measured and the pressure increase generated by the evolved hydrogen was neglected in the calculations. No hydrogen evolution was observed from sacrificial electron

donor aqueous solution under solar simulator illumination in absence of a photocatalyst.

## Kinetic H$_2$ production experiment

A flask was charged with TFP-BpyD nano-COF colloid (0.25 mL) or TFP-BD nano-COF colloid (0.5 mL), diluted hexachloroplatinic acid solution as a platinum precursor (15 % wt *vs* photocatalysts), ascorbic acid aqueous solution (0.2 M, 25 mL), and sealed with a septum. The resulting solutions were ultrasonicated for 5 min before degassing thoroughly by N$_2$ bubbling for 30 min. The reaction mixture was illuminated with a 300 W Newport Xe light-source (Model: 6258, Ozone free) for the time specified at a fixed distance under atmospheric pressure. The Xe-light source was cooled by water circulating through a metal jacket. Gas samples were taken with a gas-tight syringe and run on a Bruker 450-GC gas chromatograph equipped with a Molecular Sieve 13X 60-80 mesh 1.5 m × 1/8″ × 2 mm ss column at 50 °C with an argon flow of 40 mL min$^{-1}$. Hydrogen was detected with a thermal conductivity detector, referencing against standard gases with known concentrations of hydrogen. Hydrogen dissolved in the reaction mixture was not measured and the pressure increase generated by the evolved hydrogen was neglected in the calculations. The rates were determined from a linear regression fit and the error is given as the standard deviation of the amount of hydrogen evolved. No hydrogen evolution was observed for a mixture of H$_2$PtCl$_6$ and ascorbic acid aqueous solution under $\lambda > 295$ nm illumination in absence of a photocatalyst.

## External quantum efficiency (EQE) measurements

The external quantum efficiency for the photocatalytic H$_2$ evolution was measured using $\lambda = 420$ nm (80 mW), $\lambda = 515$ nm (45 mW) and $\lambda = 595$ nm (90 mW) LEDs controlled by an IsoTech IPS303DD power supply. For the measurement, TFP-BpyD nano-COF colloid (0.4 mL) or TFP-BD nano-COF (0.8 mL) was suspended in an aqueous solution containing ascorbic acid (0.2 M, 39.6 or 39.2 mL) with diluted H$_2$PtCl$_6$ solution as a platinum precursor (15 wt%) before illuminating with the LED. The light intensity was measured with a ThorLabs S120VC photodiode power sensor controlled by a ThorLabs PM100D Power and Energy Meter Console and the external quantum efficiency was estimated using Eq. (4):

$$EQE = 2 \times \frac{moles\ of\ hygrogen\ evolved}{moles\ of\ incident\ photons} \times 100\% \qquad (4)$$

## Time-correlated single photon counting (TCSPC) measurements

All TSCPC experiments were carried out for emission at 371 nm. TCSPC experiments were measured on an Edinburgh FLS1000 Photoluminescent Spectrometer equipped with Fluoracle software for data acquisition and analysis. The instrument response was measured with colloidal silica (LUDOX® HS-40, Sigma-Aldrich) at the excitation wavelength. Lifetimes were fitted in the FAST software using three exponential decay method.

## Quantum yield (QY) measurements

Samples were measured using Edinburgh FLS1000. It is a modular fluorescence spectrometer for measuring spectra from 200 to 900 nm. The system is constructed from the light source, excitation monochromator, sample chamber, emission monochromator and detector. The Xe2 is a 450 W xenon arc lamp that emits continuous radiation from 230 nm to >1000 nm. The light from the xenon arc is focused into the monochromator using a high-reflectivity off-axis ellipsoidal mirror, ensuring excellent focus at the entrance slit and completely uniform illumination of the monochromator grating. The monochromators in the FLS1000 are of Czerny-Turner configuration with 325 mm or 2 x 325 mm focal length and feature triple grating turrets with up to three gratings on each turret and computer-controlled slits. The grating turrets are micro-stepper motor driven with a minimum step size of 0.01 nm, a maximum slew rate of 250 nm/s can be achieved. Detector: PMT-900 with the wavelength coverage from 200-850 nm. It operates at the −21.8 °C.

## Data availability

All data are available in the main text or the supplementary information. Source data are provided with this paper.

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

## Acknowledgements

The authors acknowledge funding from the Leverhulme Trust via the Leverhulme Research Centre for Functional Materials Design. The authors thank Dr Yang Bai, Dr Haofan Yang and Chao Li for useful discussions. We thank the Materials Innovation Factory (MIF) team for help with instrument training. We thank the National Natural Science Foundation of China (NSFC, 22375062); Shanghai Municipal Science and Technology Major Project (2018SHZDZX03, 21JC1401700); Shanghai Municipal Science and Technology (20120710200); Shanghai Pilot Program for Basic Research (22TQ1400100-10); the Fundamental Research Funds for the Central Universities; Shanghai Pujiang Program (22PJ1402400); "Chenguang Program" supported by Shanghai Education Development Foundation and Shanghai Municipal Education Commission (22CGA32); the Engineering and Physical Sciences Research Council (EPSRC) for financial support under grants EP/N004884/1 and EP/P034497/1. AIC thanks the Royal Society for a Research Professorship (RSRP\S2\232003). W.W.Z. acknowledges the Young Elite Scientists Sponsorship Program by CAST (2023QNRC001) and the Research Center of Analysis and Test of East China University of Science and Technology (ECUST) for assistance with various characterizations. The TEM analysis was performed in the Albert Crewe Centre for Electron Microscopy, a University of Liverpool Shared Research Facility.

## Author contributions

W.Z. synthesized and characterized the materials, performed photocatalytic experiments, and analyzed the photocatalysis results. W.Z. performed the SEM, FT-IR measurements, PXRD measurements, gas adsorption, TGA and UV measurements. L.L. performed PL and UV measurements. B.L. performed supercritical CO2 drying. J.Y., M.Y., L.J.L. and Y.X. gave useful discussions. M.B. and N.D.B. performed TEM measurements. M.C. and Z.Z. performed the TAS measurements. W.W.Z., X.L. and W.H.Z. performed the cryo-TEM and STEM measurements. W.Z. conceived the modeling strategy. Data were interpreted by all authors and the manuscript was prepared by A.I.C, W.W.Z. and W.Z.

## Competing interests

The authors declare no competing interests.
