## [Peer Review File · Nature Communications]

Nanoscale Covalent Organic Frameworks for Enhanced Photocatalytic Hydrogen ProductionEditorial Note: This manuscript has been previously reviewed at another journal that is not operating a transparent peer review scheme. This document only contains reviewer comments and rebuttal letters for versions considered at *Nature Communications*.

REVIEWER COMMENTS

Reviewer #1 (Remarks to the Author):

The revised manuscript by Zhao et al. reflects improvements based on reviewers' feedback, including necessary explanations and additional characterizations like solid-state ¹³C NMR, XPS analyses, etc. These enhancements have rendered the manuscript well-organized and scientifically robust compared to the initial submission. Given its novelty and scientific content, the manuscript is deemed suitable for publication, with attention to minor concerns:

1. The N₂ adsorption and desorption profiles of TFP-BpyD nano-COF should be re-evaluated, particularly focusing on the micropore region (Supplementary Figure 9), to ensure accurate pore size analyses.
2. The manuscript should incorporate the rationale behind selecting ascorbic acid (AA) as a sacrificial electron donor. This addition will enrich the contextual understanding of the experimental approach.

Addressing these minor concerns will further strengthen the manuscript and ensure its alignment with publication standards of "Nature Communications".

Comments for Reviewer #2:

The manuscript from Zhao et al. has undergone revisions based on the feedback from the reviewers. In response to suggestions, the authors have made several improvements, including:

- Providing a list of other organic photocatalysts alongside representative pristine COFs in Table S3.
- Conducting X-ray photoelectron spectroscopy (XPS) analyses of TFP-BpyD and TFP-BD nano-COFs (Supplementary Figure 7a).
- Explaining the reason for the photoluminescence (PL) intensity quenching with increasing concentration (Supplementary Figure 47).
- Revising the captions of Supplementary Figure 46 and adding diffraction indices for each peak in Supplementary Figures 17 and 18.
- Assigning all vibration frequencies of each band in the Fourier-transform infrared spectroscopy (FT-IR) in Supplementary Figure 16.
- An explanation of the two PL emission bands is provided in Figure 3b.
- Including the average lifetime from time-correlated single-photon counting (TCSPC) experiments (Supplementary Figure 45).
- Describing the mode of operation and radius of curvature of the tip in atomic force microscopy (AFM) analyses.

- Enhancing the quality of figures and revising references.

These changes have significantly improved the quality and suitability of the manuscript for publication in Nature Communications. Thus, the manuscript could be accepted for publication in its current form.

Reviewer #2 (Remarks to the Author):

The authors submitted the revised manuscript "Nanoscale Covalent Organic Frameworks for Enhanced Photocatalytic Hydrogen Production" The reviewer believes that this manuscript is not suitable to publish in this journal.

1. The morphology of TFP-BpyD nano-COF and TFP-BD nano-COF are unclear in the TEM and AFM. In addition, all these materials are reported before.
2. There is a mistake in Figure 1.
3. the reviewer believes these two COF materials have no fluorescence properties or weak emission. how about their quantum yields?
4. The solid-state NMR, FT-IR, NMR, SEM, and BET of TFP-BpyD nano-COF and TFP-BD nano-COF after the photocatalytic reaction to further ensure its stability.
5. The manuscript title is unclear and not interesting to journal readers.
6. The reviewer didn't see any data about the apparent quantum yields for these materials and real photos of H₂ bubbles of these materials.

Reviewer #1 (Remarks to the Author):

The revised manuscript by Zhao et al. reflects improvements based on reviewers' feedback, including necessary explanations and additional characterizations like solid-state ^{13}C NMR, XPS analyses, etc. These enhancements have rendered the manuscript well-organized and scientifically robust compared to the initial submission. Given its novelty and scientific content, the manuscript is deemed suitable for publication, with attention to minor concerns:

1. The N_2 adsorption and desorption profiles of TFP-BpyD nano-COF should be re-evaluated, particularly focusing on the micropore region (Supplementary Figure 9), to ensure accurate pore size analyses.

Response: We thank the reviewer for this comment. We have now re-measured the N_2 sorption focusing on the micropore region. **Supplementary Figure 1** has been updated as follows:

Supplementary Figure 2. N_2 adsorption and desorption profiles at 77.3 K (left), pore size distribution profile calculated by DFT (center) and BET surface area plot derived from N_2 sorption isotherm of TFP-BpyD nano-COF (right).

2. The manuscript should incorporate the rationale behind selecting ascorbic acid (AA) as a sacrificial electron donor. This addition will enrich the contextual understanding of the experimental approach.

Response: We selected ascorbic acid (AA) as a sacrificial electron donor because of its strong reducibility for ketoenamine COFs, as evidenced by previous works (*Nature*, **2022**, 604, 72–79; *Nat. Commun.*, **2023**, 14, 593; *Nat. Commun.*, **2021**, 12, 3934 and *Nat. Commun.*, **2022**, 13, 5768).

Also, our previous work (*Nat. Chem.*, 2018, **10**, 1180–1189, Supplementary Table 1) with the ketoenamine COF, FS-COF, showed the highest performance using AA as sacrificial agent, while other sacrificial electron donors (Na₂S, Na₂SO₃, TEOA and TEA) gave lower performance.

Following the reviewer's suggestion, we have now provided a rationale for the selection of AA as a sacrificial electron donor in the revised manuscript as follows:

“Here, we selected ascorbic acid (AA) as a sacrificial electron donor because of its strong reducibility for ketoenamine COFs, as evidenced by previous work¹⁸”.

Addressing these minor concerns will further strengthen the manuscript and ensure its alignment with publication standards of “Nature Communications” .:

Response: We thank the reviewer again for his/her insightful comments, which further improve the manuscript.

Comments for Reviewer #2:

The manuscript from Zhao et al. has undergone revisions based on the feedback from the reviewers. In response to suggestions, the authors have made several improvements, including:

- Providing a list of other organic photocatalysts alongside representative pristine COFs in Table S3.
- Conducting X-ray photoelectron spectroscopy (XPS) analyses of TFP-BpyD and TFP-BD nano-COFs (Supplementary Figure 7a).
- Explaining the reason for the photoluminescence (PL) intensity quenching with increasing concentration (Supplementary Figure 47).
- Revising the captions of Supplementary Figure 46 and adding diffraction indices for each peak in Supplementary Figures 17 and 18.
- Assigning all vibration frequencies of each band in the Fourier-transform infrared spectroscopy (FT-IR) in Supplementary Figure 16.
- An explanation of the two PL emission bands is provided in Figure 3b.
- Including the average lifetime from time-correlated single-photon counting (TCSPC) experiments (Supplementary Figure 45).
- Describing the mode of operation and radius of curvature of the tip in atomic force microscopy (AFM) analyses.
- Enhancing the quality of figures and revising references.

These changes have significantly improved the quality and suitability of the manuscript for publication in Nature Communications. Thus, the manuscript could be accepted for publication in its current form.

Response: We thank this again for these insightful comments, which further improve the manuscript.

Reviewer #3 (Remarks to the Author):

The authors submitted the revised manuscript "Nanoscale Covalent Organic Frameworks for Enhanced Photocatalytic Hydrogen Production" The reviewer believes that this manuscript is not suitable to publish in this journal.

1. The morphology of TFP-BpyD nano-COF and TFP-BD nano-COF are unclear in the TEM and AFM. In addition, all these materials are reported before.

Response: We have now improved the AFM images in Figure 1 and added more cryo-TEM and AFM images in Supplementary Information. These images further confirm the nanoribbons (*i.e.*, flat fibers) and nanosheets (*i.e.*, disks) morphology for the TFP-BpyD and TFP-BD nano-COFs:

Figure 1. Synthesis of nano-COFs and their morphologies. The synthetic routes for TFP-BpyD and TFP-BD nano-COF (a). Cryo-TEM and AFM images of TFP-BpyD nano-COF (b, c) and TFP-BD nano-COF (d, e).

Supplementary Figure 3. Cryo-TEM images of TFP-BpyD nano-COF (a,) and TFP-BD nano-COF (b).

Supplementary Figure 4. AFM images of TFP-BpyD nano-COF (a) and TFP-BD nano-COF (b).

In addition, all these materials are reported before.

The reviewer pointed this out before. We would again reiterate that while the basic chemical structure has been reported, TFP-BpyD COF and TFP-BD COF were studied previously as bulk solids. Here we synthesized the TFP-BpyD and TFP-BD nano-COF at the *nanoscale*, and the materials are clearly distinct in their properties with respect to the previously reported bulk materials (*Angew. Chem. Int. Ed.* 2022, 61, e202200413).

As we have addressed in the first revision, most research efforts so far for COFs have focused on chemical diversity, while the nano-morphology is often overlooked. This is true not only in

photocatalysis, but also many other optoelectronic applications. This point has also been acknowledged recently by Rahul Banerjee, an innovator and leader in the field of COFs (*ACS Nano* 2021, 15, 12723–12740; *J. Am. Chem. Soc.* 2022, 144, 11482–11498):

“However, even after a decade and a half of research, the primary focus of the field lies on the covalent reticular structure, and the chemistry of COFs in the nano regime is often overlooked. The COF nanostructures (nanosheets, nanofibers, and nanospheres) are quite distinct from the bulk forms and often influence the properties of their bulk counterparts.”

Here, we show that the size effect of COFs can dramatically affect their photocatalytic performances; indeed, to a greater degree than making small chemical modifications to the base COF material (*i.e.*, yet another ‘new’ chemical structure). Nano-sized COFs give rise to greatly improved light-harvesting and charge carrier generation compared to their bulk counterparts, even though both have the same nominal chemical structure. This results in greatly enhanced sacrificial hydrogen evolution rate (HERs) of 392.0 mmol g⁻¹ h⁻¹ for TFP-BpyD nano-COF, which is among the highest reported for all organic materials.

The significance of this work extends beyond the synthesis of nano-sized COFs and demonstration of photocatalytic activity. For the first time, we demonstrate a reverse concentration-dependent photocatalytic phenomenon for nano-sized COFs, whereby a higher photocatalytic activity was found at a lower catalyst concentration. We studied this phenomenon by photoluminescence and transient absorption spectroscopy and found substantial singlet annihilation in nano-COFs under higher particle concentration. These findings may be of broad importance for understanding the underpinning photophysical process for COF photocatalysts. As in the first revision, we would respectfully suggest that the field at this stage needs more basic understanding of this type, rather than more and more ‘novel’ synthetic examples with no in-depth studies of their properties or the influence of nanoscale morphology.

2. There is a mistake in Figure 1.

Response: We thank the reviewer for spotting this. This was a typographical error and we have revised the Figure 1 as follows:

Figure 1. Synthesis of nano-COFs and their morphologies. The synthetic routes for TFP-BpyD and TFP-BD nano-COF (a). Cryo-TEM and AFM images of TFP-BpyD nano-COF (b, c) and TFP-BD nano-COF (d, e).

3. the reviewer believes these two COF materials have no fluorescence properties or weak emission. how about their quantum yields?

Response: We have now provided the quantum yields of TFP-BpyD nano-COF at different concentrations, which also followed the reverse concentration-dependent trend. At lower concentration (0.1 mL), the quantum yield was 0.17 %. With the increase of concentration of TFP-BpyD nano-COFs, the quantum yields decreased to 0.13 % (0.5 mL) and 0.10 % (1.0 mL).

These data have been added in the Supplementary Information as follows:

Supplementary Table 4. Quantum yields analysis of TFP-BpyD nano-COF.

TFP-BpyD nano-COF	Water	Quantum yields	Average quantum yields
0.1 mL	4.9 mL	Test 1	0.16%
		Test 2	0.18%
0.5 mL	4.5 mL	Test 1	0.12%
		Test 2	0.13%
1.0 mL	4.0 mL	Test 1	0.10%
		Test 2	0.10%

Note: Quantum yields of TFP-BpyD nano-COF at different concentrations followed a reverse concentration-dependent phenomenon as well. At lower concentration (0.1 mL), the quantum yield is 0.17 %. With the increase of concentration of TFP-BpyD nano-COF, the quantum yields decreased to 0.13 % (0.5 mL) and 0.10 % (1.0 mL).

4. The solid-state NMR, FT-IR, NMR, SEM, and BET of TFP-BpyD nano-COF and TFP-BD nano-COF after the photocatalytic reaction to further ensure its stability.

Response: In our previous version of manuscript, we measured the FT-IR and TEM for nano-COFs after photocatalysis. FTIR characterization showed no significant changes for post-photocatalytic materials (**Supplementary Figure 42**), indicating the retained chemical integrity. TEM images suggested retention of the nanoscale morphology for nano-COFs with Pt nanoparticle uniformly distributed on its surface, though some aggregations of COF nanoparticles were observed (**Supplementary Figures 38–39**). HAADF-STEM images and elemental mappings (**Supplementary Figures 40 and 41**) also confirmed the even dispersion of Pt cocatalyst on the nano-COFs.

Following the reviewer's suggestion, we conducted additional NMR and SEM measurement for nano-COF before and after photocatalysis. No monomer signals were detected in the NMR analysis. These results provide further evidence of the chemical stability of these nano-COF photocatalysts; that is, no soluble, NMR-detectable organic fragments were formed.

Supplementary Figure 5. NMR spectra of TFP-BpyD nano-COF (**a**) and TFP-BD nano-COF (**b**) before and after photocatalysis. No monomer signals were detected. The shift for water signals (~3.33 ppm) after photocatalysis was due to the presence of AA.

Supplementary Figure 6. SEM images of TFP-BpyD nano-COF (**a**) and TFP-BD nano-COF (**b**) after photocatalysis.

We have now added the NMR and SEM data after photocatalysis in the revised manuscript as follows:

“After photocatalysis, the TFP-BpyD and TFP-BD nano-COF materials were characterized by TEM, STEM, SEM, NMR and FT-IR.”

“FTIR and NMR characterizations indicated no significant changes after photocatalysis (**Supplementary Figures 42 and 43**), while TEM and SEM (**Supplementary Figure 44**) analysis suggested some aggregation of the COF crystallites”.

For solid-state NMR and N₂ adsorption (for BET surface areas) tests, typically at least 100 mg of solid-state COFs are required for both measurements. However, in our nano-COFs systems, only 0.343 mg nano-COF was used in 100 ml water for photocatalysis. To acquire adequate COFs for solid-state NMR and N₂ adsorption tests, a total volume of 30 liters of photocatalytic system would need to be devised and executed. Given the high dispersibility of nano-COF in aqueous solution, gathering such a substantial sample would be challenging. We believe that the characterization of FT-IR, NMR, SEM and TEM have collectively validated the chemical stability of the nano-COFs.

5. The manuscript title is unclear and not interesting to journal readers.

Response: As we have addressed in Q1, we aim to emphasize the nanoscale nature of these photocatalytic COFs, distinguishing them from conventional bulk materials. We feel that the title captures this, but we would welcome the Editor's view on this.

6. The reviewer didn't see any data about the apparent quantum yields for these materials and real photos of H₂ bubbles of these materials.

Response: We have provided the external quantum efficiency (EQE) in our first submission, which is same to apparent quantum yield (AQY) in photocatalytic H₂ production (*Nat Catal*, 2020, **3**,

649-655,
$$\text{AQY (\%)} = \frac{(\text{Number of evolved H}_2 \text{ molecules} \times 2)}{(\text{Number of incident photons})} \times 100\%$$
; *Nature*, 2020, **581**, 411–414,

$$\text{EQE(\%)} = 2 \times \frac{N(\text{H}_2)}{N(\text{photons})} \times 100$$
).

Relevant results can be found in Figure 2f, and Supplementary Figure 37 and Table 3.

As for optical photos of H₂ bubbles of nano-COFs, following the reviewer's suggestion, we have now provided this in the revised manuscript as follows (bubbles are indeed observed):

Supplementary Figure 7. Optical image of TFP-BpyD nano-COF for photocatalytic H₂ production.

REVIEWERS' COMMENTS

Reviewer #1 (Remarks to the Author):

The manuscript by Zhao et al. reporting the synthesis of nanoscale Covalent Organic Frameworks and their applications for photocatalytic water splitting has been revised to address all concerns and comments. Therefore, the manuscript in its current form could be accepted for publication in 'Nature Communications'.

Reviewer #2 (Remarks to the Author):

The authors answered all reviewer's comments in this revised manuscript. Therefore, this manuscript could be published in this journal.

Reviewer #1 (Remarks to the Author):

The manuscript by Zhao et al. reporting the synthesis of nanoscale Covalent Organic Frameworks and their applications for photocatalytic water splitting has been revised to address all concerns and comments. Therefore, the manuscript in its current form could be accepted for publication in 'Nature Communications'.

Response: We thank the reviewer for his/her insightful comments, which further improve the manuscript.

Reviewer #2 (Remarks to the Author):

The authors answered all reviewer's comments in this revised manuscript. Therefore, this manuscript could be published in this journal.

Response: We thank the reviewer for his/her insightful comments, which further improve the manuscript.